# Canine Gallbladder Erosion/Ulcer and Hemocholecyst: Clinicopathological Characteristics of 14 Cases

**DOI:** 10.3390/ani13213335

**Published:** 2023-10-26

**Authors:** Ikki Mitsui, Kazuyuki Uchida

**Affiliations:** 1Laboratory of Veterinary Anatomy, Faculty of Veterinary Medicine, Okayama University of Science, Imabari 794-8555, Japan; 2Laboratory of Veterinary Pathology, Graduate School of Agriculture and Life Sciences, The University of Tokyo, Bunkyo-ku, Tokyo 113-8657, Japan; auchidak@g.ecc.u-tokyo.ac.jp

**Keywords:** gallbladder, erosion, ulcer, dog, COX-1, COX-2, cholecystitis, mucus, mucocele, hemocholecyst

## Abstract

**Simple Summary:**

This study illustrates so far under-reported canine gallbladder erosion/ulcer. Based on the clinicopathological information in addition to the COX-1 and COX-2 IHC results, canine gallbladder erosion/ulcer is possibly related to decreased cytoprotection physiologically provided by arachidonic acid.

**Abstract:**

(1) Background: Gallbladder mucosal erosion and/or ulceration are illnesses associated with unexpected gallbladder intra-cystic bleeding (hemocholecyst), an under-reported problem in dogs. (2) Methods: Clinicopathological characteristics of 14 dogs with gallbladder erosion/ulcer were investigated in this single-center retrospective study using clinical data and archived gallbladder tissues of client-owned dogs. (3) Results: Canine gallbladder erosion/ulcer tends to occur in older, neutered dogs of various breeds. Vomiting, lethargy, and anorexia are common. Concurrent gallbladder rupture occurred in 5/14 cases (35.7%), while rupture was absent in 6/14 cases (42.8%) and undetermined in 3/14 (21.4%) cases. Histologically, the gallbladder wall was markedly thickened due to mucosal hyperplasia, inflammatory infiltrates, fibrosis, edema, hemorrhage, and smooth muscle hyperplasia/hypertrophy. Twelve out of fourteen cases (85.7%) had concurrent cholecystitis of varying severity. Bacteria were detected by Giemsa or Warthin–Starry stain in 8/14 (57.1%) cases. Bacterial rods immunoreactive to the anti-*Helicobacter* antibody were present in one case. Mucosal epithelial cells of the gallbladder erosion/ulcer cohort were immunopositive for the cyclooxygenases COX-1 or COX-2 in only 5/14 (35.7%) cases. In contrast, COX-1 and COX-2 were more frequently expressed in a reference pool of cases of gallbladder mucocele (*n* = 5) and chronic cholecystitis (*n* = 5). COX-1 was expressed in 9/10 cases (90.0%) of gallbladder mucocele and chronic cholecystitis and in 10/10 cases (100%) for COX-2. (4) Conclusions: Canine gallbladder erosion/ulcer is an under-reported condition which requires active clinical intervention. Based on the clinicopathological information reported in this study in addition to the COX-1 and COX-2 IHC results, we suggest that canine gallbladder erosion/ulcer may be related to decreased cytoprotection physiologically provided by arachidonic acid, but which is decreased or absent due to reduced COX expression because of yet undetermined etiologies.

## 1. Introduction

The gallbladder is a saccular juxtahepatic organ that accumulates and empties bile in response to physiological hormonal and nerve inputs [1,2]. Unlike human gallbladders, canine gallbladders are seldom afflicted with gallstones [3] or cancer. Instead, acute bacterial cholecystitis, gallbladder rupture, gallbladder mucocele, and chronic cholecystitis are more frequently reported diseases in canine patients [4,5,6,7,8,9,10,11,12,13,14].

Gallbladder erosion and/or ulcer is a superficial or full-thickness loss of the gallbladder mucosa inevitably accompanied by intraluminal hemorrhage (hemocholecyst, defined as non-traumatic intraluminal hemorrhage of the gallbladder [15]), and, when hemorrhage is severe enough, acute to subacute severe clinical signs ensue. Considering these features, canine cases with gallbladder erosion, ulcer, and hemocholecyst may also be reported as hemorrhagic or necrotizing cholecystitis. A literature search of online databases (Google Scholar and National Centre for Biotechnology Information) for the various combinations of key words “gallbladder”, “ulcer”, “necrotizing”, “hemorrhagic”, “cholecystitis”, “dog”, “hemocholecyst”, and “hemobilia (defined as bleeding into the biliary tree [16])” resulted in no relevant matches as of 6 October 2023 except for a case report on canine acute hemobilia and hemocholecyst associated with gallbladder carcinoid [17]. Although a case series of canine gallbladder infarct has been reported, its description of a full-thickness necrosis of the gallbladder wall differs from the description of gallbladder erosion/ulcer in which erosion/necrosis is confined to the mucosa in our study [18].

There are a few case reports and case series on human cases of hemorrhagic or necrotizing cholecystitis with hemocholecyst/hemobilia [19,20,21]. Possible etiological factors include iatrogenic causes, cholelithiasis, trauma, neoplasm, bleeding diathesis including anticoagulant medication, and vasculitides [20,21]. In addition, erosion or ulcers in human gallbladders have been described as a non-core histological finding in cases of acute/chronic calculous cholecystitis, immunoglobulin G4-associated cholecystitis, xanthogranulomatous cholecystitis, and ischemic gallbladder disease [22]. Since canine gallbladders are seldom affected by potentially tissue-damaging gall stones, understanding the etiopathogenesis of canine gallbladder erosion/ulcers would aid in the future development of effective preventative and therapeutic methods for gallbladder erosion/ulceration with resultant gallbladder hemocholecyst/hemobilia.

Arachidonic acid is an anti-inflammatory agent which is converted to prostaglandins by cyclooxygenases (COX). Prostaglandin E_2_ (PGE_2_) is known to induce mucin secretion by gallbladder mucosal epithelial cells of cats [23] and dogs [24,25] in experimental settings. Nilsson and colleagues demonstrated increased COX-2 expression in experimentally induced inflamed feline gallbladders using immunoblotting [26]. They further experimentally demonstrated that the administration of COX-2 inhibitors reduced PGE_2_ production in feline gallbladders, highlighting the usefulness of COX-2 inhibitors as a therapeutic option for human chronic cholecystitis patients awaiting cholecystectomy instead of non-steroidal anti-inflammatory drugs (NSAIDs) [26]. Obstruction caused by excessively accumulated mucin in the inflamed gallbladder has been suspected to be the source of biliary pain [27], so therapy for chronic cholecystitis through decreasing mucin secretion by COX-2 inhibitors seems promising. COX-2 inhibitor therapy is also considered appropriate for acute cholecystitis based on experiments by Longo and colleagues using cultured human gallbladder mucosal epithelial cells [28]. Regarding COX-1, its cytoprotective role on the gastrointestinal mucosa is well known [29]. Accordingly, a balance between a detrimental and protective effect of cyclooxygenases has been the subject of research and pharmaceutics. Investigating expression patterns of COX-1 and COX-2 in naturally diseased canine gallbladders has not yet been undertaken but is essential to understand the pathogenesis of canine gallbladder diseases that tend to have unique combinations of hyperplasia, inflammation, mucus hypersecretion, mucocele, and mucosal erosion/ulcer, all of which are related to mucosal epithelial cell morphology and functions.

The aim of this study was two-fold: 1. to describe clinicopathological characteristics of canine cases with gallbladder erosion/ulcer, and 2. to compare these cases with those with gallbladder mucocele or chronic cholecystitis to gain insight on the etiopathogenesis of canine gallbladder erosion/ulcer.

## 2. Materials and Methods

### 2.1. Cases

The data of client-owned dogs, whose surgically excised gallbladders had been submitted to No Boundaries Animal Pathology, LLC (affiliated with Ikki Mitsui, Tokyo, Japan) between January 2016 and July 2019, were analyzed for the presence of (Japanese) terms such as “gallbladder”, “erosion”, “ulcer”, and “hemorrhage” in histopathology reports. The term “hemocholecyst” or “hemobilia” was not searched because it had never been used in the authors’ histopathology report until the beginning of this study. Fourteen cases (gallbladder ulcer, GU) fulfilled the above criterion and were further examined. In addition, 10 cases were arbitrarily chosen, including 5 cases that had a typical histology of mucocele (gallbladder mucocele, GM) plus 5 cases in which gallbladders histologically showed severe lymphoplasmacytic (chronic) cholecystitis (chronic cholecystitis, CC), and were examined in the same way for comparison. Verbal and/or written consent was obtained from the owners of the dogs before surgery by the attending veterinarians in all 24 cases. Information on the subjects is summarized in Appendix A. Signalment, clinical information, and ultrasonographic descriptions were collected from the description on the case submission sheets. Cholecystocentesis was not performed in any cases in this study. Since no live research animals were used in our study, approval by the Institutional Animal Care and Use Committee (IACUC) was waived by the Clinical Research Ethics Committee of the Faculty of Veterinary Medicine of the Okayama University of Science. The gallbladder of a 4-year-old, intact male Beagle dog was used as a clinical control sample. The dog showed no clinical abnormality associated with the gallbladder and liver. The Beagle was used for urological research and its usage was approved by the IACUC of the Okayama University of Science (Approval No. 2023-034).

### 2.2. Histopathology

All gallbladder samples were fixed in 10% neutral-buffered formalin immediately after cholecystectomy. After the clinical control dog was humanely euthanized by potassium chloride intravenous injection under anesthesia by intravenous propofol injection (6 mg/kg), the gallbladder was immediately collected and fixed in the same manner. These gallbladders were trimmed at multiple points at right angles to the long axis of the gallbladder. The samples (multiple tissues per case) were embedded in paraffin after 1 to 3 days of fixation. Four-micrometer-thick sections were stained with hematoxylin and eosin (H&E) for histopathological examination by two board-certified veterinary pathologists (I.M., DACVP (anatomic)/DJCVP, and K.U., DJCVP). The criteria for examination are listed in Appendix A. Gallbladder mucosal inflammation was graded histologically based on the published criteria [11]: severe, mild, or no inflammation. Mucocele was characterized histologically by abundant, amorphous, amphophilic, thick gelatinous material expanding the gallbladder lumen and adhered to the hyperplastic mucosa. Gallbladder wall total thickness (GWTT) was measured on the whole-slide images, prepared by a virtual slide scanner (NanoZoomer S210, Hamamatsu Photonics K.K., Shizuoka, Japan). GWTT measurement was defined as the distance between mucosal and serosal surfaces, avoiding extreme thickness or thinness in each specimen [11].

### 2.3. Histochemistry

Four-micrometer-thick sections of the gallbladders of 24 dogs belonging to GU, GM, and CC groups and the gallbladder of the clinical control dog were stained with Giemsa, Warthin–Starry (WS), and phosphotungstic acid hematoxylin (PTAH) following the established protocols. The PTAH stain was omitted for the clinical control sample because of an obvious lack of thrombi during routine histological examination. As for the high iron diamine-alcian blue (HID-AB) pH 2.5 method, performed herein to characterize the chemical features of mucin, sections were placed in the HID solution for 16 h at room temperature, rinsed with distilled water and 3% acetic acid solution, placed in an AB pH 2.5 solution for 30 min, rinsed with 3% acetic acid and distilled water, counterstained by Kernechtrot solution, dehydrated, and mounted. Normal canine esophageal glands were used as a control for the HID-AB pH 2.5 method. Sections stained by Giemsa or Warthin–Starry were examined for bacterial pathogens, and the presence and morphology of bacteria were recorded. PTAH was used to detect fibrin thrombi. Sections stained with HID-AB pH2.5 were evaluated for the presence of sulfomucins (black staining) or sialomucins (blue staining). Results of the HID-AB pH2.5 stain were documented according to the dominant mucus type (sulfomucin vs. sialomucin). In the case that both mucus types were equally present, they were recorded as “mixed mucus type”. If the mucus types of mucosal surfaces and crypts stained differently, they were recorded as “two-tones”.

### 2.4. Immunohistochemistry

Immunohistochemistry (IHC) was performed for gallbladder specimens of all cases including the clinical control dog to observe the immunoreactivity of gallbladder mucosal epithelial cells to anti-cyclooxygenase-1 (COX-1) and anti-COX-2 antibodies. In addition, an antibody to *Helicobacter* was used to examine the presence of this bacterial species in all gallbladder samples including that of the clinical control dog. Briefly, four-micrometer-thick sections were placed and dried on Crest slide glasses (Matsunami Glass Ind., Ltd., Osaka, Japan). Table 1 shows primary antibodies used for IHC, their host, type, dilution, source, and catalogue number. Slides were baked at 60 °C for 30 min then deparaffinized in xylene and rehydrated in graded alcohol solutions and distilled water. Antigen retrieval was performed by heating slides in a pressure cooker at 117 °C for 10 min in pH 6.0 citrate buffer. Endogenous peroxidase was inhibited by immersion in 3% H_2_O_2_ in distilled water for 10 min. Non-specific immunoreaction was blocked by incubating tissue slides with 2.5% normal horse serum (R.T.U. Universal Elite ABC Kit, Catalog No. PK-7200, Vector Laboratories Inc., Burlingame, CA, USA) for 20 min at room temperature. Conjugation with primary antibodies was conducted at 4 °C overnight. Reaction with a secondary antibody as well as a biotinylated universal antibody was conducted following the manufacturer’s instructions (R.T.U. Universal Elite ABC Kit, Catalog No. PK-7200, Vector Laboratories Inc., Burlingame, CA, USA). Immunoreaction was visualized by applying a diaminobenzidine solution (ImmPACT DAB Peroxidase Substrate Kit, Catalog No. SK-4105, Vector Laboratories Inc., Burlingame, CA, USA) and briefly counterstaining with hematoxylin. As a positive control, the following slides were simultaneously stained: tissue sections of canine normal stomach for COX-1 IHC; tissue sections of canine mammary simple carcinoma for COX-2 IHC; and endoscopic tissue specimens of canine stomach with spirochetes for *Helicobacter* IHC. As a negative control, the same IHC procedure was conducted using non-immune rabbit IgG (RABBIT IgG, Catalog No. I-1000, Vector Laboratories Inc., Burlingame, CA, USA) at the manufacturer’s recommended dilution (0.1 µg/mL) as well as PBS instead of a primary antibody. The IHC of all positive- and negative-control specimens validated the above-mentioned IHC procedures. The stained IHC slides were examined independently by the authors. The immunohistochemical staining was judged positive when more than 50% of gallbladder epithelial cells were intensely positive.

## 3. Results

### 3.1. Signalment, Duration between Clinical Onset and Cholecystectomy, Presence/Absence of Rupture, and Chief Complaint

The average age of the gallbladder erosion/ulcer (GU) cases was 122.6 months (about 10.2 years). The median age of the GU cases was 116 months (about 9.6 years), while the range was from 69 to 170 months (about 5.7 to 14.1 years). Mixed breeds (*n* = 4) and Miniature Dachshunds (*n* = 3) were more common among the GU patients. There was one each of Toy Poodle, Chihuahua, Shiba Inu, Yorkshire Terrier, Shetland Sheepdog, Pembroke Welsh Corgi, and German Shepherd. There were six castrated males (42.8%), five spayed females (35.7%), two intact males (14.2%), and one intact female (7.1%) in the GU cohort. The average duration between onset of clinical signs and open cholecystectomy was 7.35 days (range: 0 to >30 days) among the GU patients. Gallbladder rupture was seen in five dogs, absent in six dogs, and unknown in three dogs in the GU cohort, while the gallbladder mucocele (GM) and chronic cholecystitis (CC) groups lacked evidence of rupture. Vomiting (*n* = 8, 57.1%), lethargy (*n* = 7, 50.0%), and decreased appetite (anorexia; *n* = 6, 42.8%) were common among the GU patients. The results of the five cases of GM and five cases of CC regarding signalment, duration between clinical onset and cholecystectomy, presence/absence of rupture, and chief complaint are described in Appendix A.

### 3.2. Ultrasonography, Complete Blood Count, Blood Chemistry, and Lesions in Other Organs

Ultrasonographic gallbladder abnormalities of the GU group included thickened wall (*n* = 3), distention (*n* = 2), sludge (*n* = 2), and one each of mucocele, gallstone, peritonitis, contents with shadowing, immovable contents other than mucocele, ascites, and rupture. The details of ultrasonographic abnormalities of three dogs were not provided by the submitters. Complete blood count (CBC) values were unremarkable in eight cases of the GU cohort. Leukocytosis with (*n* = 2) or without (*n* = 1) left shift were detected in three patients in the GU group. Mild anemia was noticed in two cases in the GU group. The results of CBC of one dog in the GU group were not provided by the submitter. Twelve dogs of the GU cohort (12/14, 85.7%) had abnormal blood chemistry values indicative of hepatobiliary disease; however, most of the provided data lacked sufficient details such as date of examination and/or reference intervals. Lesions in other organs in the GU cohort included common bile duct occlusion (*n* = 2) and one each of small intestinal lymphatic dilation, splenic nodular mass, hepatic nodular mass, and microhepatia. The results of the five cases of GM and five cases of CC regarding ultrasonography, complete blood count, blood chemistry, and lesions in other organs are described in Appendix A.

### 3.3. Histologic Findings

All GU cases had mucosal erosion and/or ulceration of various geographical degree (focal to extensive) with hemorrhage. Gross appearance of the lesions is shown in Appendix A. Histologic lesions are depicted in Figure 1a–n. Gallbladder mucosal epithelial cells at the rim of erosion/ulceration generally showed no significant morphological change (Figure 2a) but with rare pyknosis. Histological characteristics other than erosion/ulceration markedly varied among 14 cases. Histological gallbladder contents of the dogs of the GU cohort included blood (*n* = 13; 92.8%), mucus (*n* = 9; 64.2%), bile (*n* = 5; 35.7%), suppurative exudate (*n* = 3; 21.4%), and fibrin (*n* = 2; 14.2%). Twelve out of fourteen cases of the GU cohort had infiltration of lymphocytes and plasma cells in the lamina propria (chronic cholecystitis), with ten cases (71.4%) being judged severe (Figure 2b,c) and the remaining two (14.2%) as mild degree. Two cases had mucocele with or without concurrent chronic cholecystitis (Figure 2d). Three cases (GU5, 13, and 14) of the twelve GU cases with chronic cholecystitis had severe infiltration of neutrophils with hemorrhage and fibrin exudation confined to the ulcerated mucosa and submucosa (Figure 2e,f). The average gallbladder wall total thickness (GWTT) in GU patients was 1685 µm (range 840–3800 µm), i.e., approximately 3 times thicker than that of the clinical control dog (550 µm) due to fibrosis, smooth muscle hyperplasia/hypertrophy, interstitial edema, and mixed inflammatory infiltrates (Figure 1o,p). The average GWTT of CC and GM patients were 1430.4 µm (range 912–1780 µm) and 1574.2 µm (range 481–2710 µm), respectively. The GM cohort was histologically characterized by a luminal distention by abundant semi-solid gelatinous contents and mucosal hyperplasia (Figure 1q–u), and they tended to have concurrent minimal/mild cholecystitis and variably increased GWTT (up to 5 times thicker than the normal canine gallbladders). The CC cohort was characterized by severe cholecystitis and variably increased GWTT (2 to 3 times thicker than the normal canine gallbladders) (Figure 1v–z). No fibrin thrombi were detected by PTAH in any of 24 samples.

### 3.4. Bacterial Detection and Chemical Property of Mucus

The results are summarized in Table 2. Bacteria were demonstrated by Giemsa and Warthin–Starry (WS) stains in 8/14 (57.1%) GU, 5/5 (100%) GM, and 0/5 (0%) CC cases (Figure 3a–e). The gallbladder of the clinical control dog did not reveal bacteria by either staining method. The morphology of bacteria varied from cocci to coccobacilli to rods. Bacterial rods were well delineated by the WS stain, but bacteria of typical spirochete morphology were not detected in any samples stained with WS. Rod-shaped bacteria of GM4 (1/24, 4.1%) were immunopositive to the anti-*Helicobacter* antibody (Figure 3f). Using the HID-AB pH 2.5 method, the types of mucus produced by the gallbladder mucosal epithelial cells were found to vary markedly among individual cases (Figure 3g–l). In the GU group, there were five cases of mixed pattern, five cases of a sulfomucin-dominant pattern, two cases of two-tone mucin production (sulfomucin by mucosal surface and sialomucin by crypts), one case of a sialomucin-dominant pattern, and one case of unstained mucus. In the GM group, two cases were of a sulfomucin-dominant pattern, two cases were mixed pattern, and there was one sialomucin-dominant pattern. In the CC group, three cases showed a mixed pattern while two cases were a sulfomucin-dominant pattern.

### 3.5. Expression of COX-1 and COX-2

The results are summarized in Table 2. Expression of COX-1 (5/14; 35.7%) and COX-2 (5/14; 35.7%) by gallbladder mucosal epithelial cells was demonstrated by IHC in the GU cohort (Figure 4a–n and Figure 5a–n). In contrast, four out of five (80.0%) cases in the GM cohort were COX-1 positive and all (100%) cases were COX-2 positive (Figure 4q–u and Figure 5q–u). In the CC group, all (100%) cases had positive immunoreaction to both anti-COX-1 and anti-COX-2 antibodies (Figure 4v–z and Figure 5v–z). The gallbladder mucosa of the clinical control dog was immunonegative for COX-1 and COX-2 antibodies (Figure 4o and Figure 5o). Positive IHC control samples showed the expected reaction for the COX-1 or COX-2 antibody (Figure 4p and Figure 5p).

## 4. Discussion

The present study describes the clinicopathological, histopathological, histochemical, and immunohistochemical features of gallbladder erosion/ulcer of dogs. Considering the frequent coexistence of a relatively short period of clinical course and chronic cholecystitis, canine gallbladder erosion/ulcer likely represents a condition known as “acute on chronic cholecystitis”, as reported in human medicine [30,31]. This condition and terminology, however, are still poorly defined, and the usage of these terms does not seem to have gained consensus among hepatobiliary experts both in human and veterinary medical fields. Still, veterinarians should be aware of and prepared for a sudden exacerbation of long-stable canine gallbladder diseases with minimal or no premonitory signs, especially in older dogs.

Neutered animals predominated in the cohort of canine gallbladder erosion/ulcer. In Japan, the neutering rate of dogs is approximately 50% according to the online source [32]. A skewed sex distribution, e.g., the predominance of neutered patients in a patient pool, has been reported for various canine gallbladder diseases, such as cholangitis, cholangiohepatitis, cholecystitis, and gallbladder rupture [4,5,11,33]. A possible explanation for these phenomena includes the derangement of sex hormones or altered metabolism; however, further investigation plus an increase in sample size are warranted to understand the exact cause of skewed sex distribution in canine gallbladder erosion/ulcer.

Gallbladder rupture was observed in five cases in our canine gallbladder erosion/ulcer cohort, and three out of the five cases had histologic evidence of mucocele. Gallbladder mucocele is known to cause gallbladder rupture due to an elevated intraluminal pressure by a continuous accumulation of immovable thick gelatinous mucus [5,9]. The remaining two out of the five cases with ruptured gallbladder had severe cholecystitis, and thus a loss of tissue integrity and decreased resistance to an elevated tension in the gallbladder wall might play a role in the process of rupture. Though gallbladder erosion/ulcer might have been perceived by pathologists as a subtype of gallbladder infarction [18], there are some important morphological differences between gallbladder infarction and erosion/ulcer. Namely, gallbladder infarction manifests as transmural gallbladder necrosis without significant inflammatory infiltrates. In addition, 3 out of the 12 reported gallbladder infarction cases had thrombi and atheromatous changes in blood vessels of the gallbladder wall, suggesting underlying vascular pathology [18]. In our cases of gallbladder erosion/ulcer, only the gallbladder mucosa was damaged, and there was no transmural necrosis or vascular wall lesions. In addition, there was significant mucosal inflammation in almost all cases (12/14; 85.7%) in our cohort of canine gallbladder erosion/ulcer. These morphological differences between the reported gallbladder infarction and our canine gallbladder erosion/ulcer cases motivated us to consider etiologic infectious agents and the expression of COX-1 and COX-2 by gallbladder mucosa epithelial cells as a possible cause of canine gallbladder erosion/ulcer.

Non-specific clinical signs such as vomiting, lethargy, and decreased appetite were common among the patients with gallbladder erosion/ulcer in our study. This showed the importance of comprehensive clinical workup including abdominal ultrasonography when evaluating canine patients with obvious but non-specific clinical illness. Ultrasonography is recommended for the evaluation of gallbladder disorders in both human and veterinary medicine because of the ease in detecting abnormalities [1,34,35]. Ultrasonography is especially useful for the evaluation of gallbladder contents, the thickness and integrity of the gallbladder wall (presence/absence of rupture), and the detection of other abnormalities in the biliary tree as well as peritonitis. It would be of value to determine if there are any specific ultrasonographic findings directly related to gallbladder erosion/ulcer and resultant intraluminal hemorrhage, but such characteristic findings have not yet been reported.

A detailed histological investigation of mucosal, focal to extensive erosion/ulcer gave little indication as to etiology in the present study. Erosion/ulcer seemed abrupt, leaving the adjacent mucosal epithelial cells almost intact. In addition, there was no “hard stuff” to cause mechanical mucosal injury such as gallstones in the canine gallbladders we examined except for one case. Bacteria were detected by histochemistry (Giemsa and Warthin–Starry stains) and immunohistochemistry (using an anti-*Helicobacter* antibody) in our gallbladder erosion/ulcer cohort as well as in the gallbladder mucocele and chronic cholecystitis cohort. Bacterial burden (number) and morphological features of the bacteria differed significantly among the cases, thus making it impossible to determine the role of bacteria in the development of gallbladder erosion/ulcer.

Erosion/ulcer of the gallbladder mucosa has been described in a variety of human conditions such as acute and chronic calculous cholecystitis, cytomegalovirus infection, IgG4-associated cholecystitis, and xanthogranulomatous cholecystitis [22], though erosion/ulcer does not seem to be the principal or pathognomonic feature in these conditions. Possible synonyms for gallbladder erosion/ulcer include hemorrhagic cholecystitis, necrotizing cholecystitis, or acute exacerbation of chronic cholecystitis, though there are no such subheadings in the established textbook chapter on gallbladder diseases [1,22,34]. A review paper on human hemorrhagic cholecystitis reports that this is a rare disease and that its pathogenesis is not fully understood, though it is thought that perhaps transmural inflammation leads to ischemia and erosion of the gallbladder mucosa with eventual gallbladder hemorrhage [20]. Administration of anti-coagulation drugs such as heparin or warfarin is mentioned as a possible trigger of human hemorrhagic cholecystitis [20]; however, there was no record of administration of these drugs in the case submission sheets of our canine gallbladder erosion/ulcer cohort. Another case series on human hemorrhagic cholecystitis reported that the etiology of two clinical cases of hemorrhagic cholecystitis was unknown or due to possible blunt trauma, respectively [19]. The aforementioned literature on human hemorrhagic cholecystitis noted that clinical diagnosis was difficult because imaging modalities could not reliably distinguish intraluminal blood (hemocholecyst) from other possible entities [19,20]. The same dilemma may apply to veterinary cases because there was no indication of hemocholecyst upon ultrasonographic evaluation in our cohort of canine gallbladder erosion/ulcer. Other than hemorrhagic cholecystitis, human literature dealing with necrotizing cholecystitis or acute exacerbation of chronic cholecystitis is sparse and lacks sufficient detail on histomorphological characteristics [36,37].

Mucus hypersecretion is a key finding in canine mucoceles [9,38,39]. Though the physiopathological significance of this mucus hypersecretion is still unclear, it could be a physiologic reaction to clearly harmful agents such as toxins, pathogens, and/or lithogenic substances. Gallbladder obstruction due to excessive mucus secretion is speculated to be the cause of biliary pain [27], and this could explain indefinite symptoms in patients with non-neoplastic gallbladder disorders. The exact mechanism by which mucus hypersecretion occurs is currently unknown. It has been shown that mucus secretion by gallbladder mucosal epithelium can be experimentally induced in cats and cultured canine gallbladder mucosal epithelial cells upon stimulation by PGE_2_ [23,25]. The administration of lipopolysaccharide also increased mucus production by cultured canine gallbladder mucosal epithelial cells [24]. It is essential to determine the precise mechanism of mucus hypersecretion in diseased gallbladders. As for treatment, based on the authors’ personal communication with veterinarians in Japan, cholecystectomy dramatically improved appetite and the general condition of dogs with long-lasting gallbladder sludge.

In general, COX-1 is constitutively expressed in most tissues rendering homeostatic functions such as cytoprotection in the gastrointestinal tract [29]. COX-1 is also induced by inflammatory stimuli [29]. COX-2 expression, on the other hand, is mainly confined to cells involving inflammatory reactions [29]. Our investigation on canine gallbladder erosion/ulcer cases revealed a diminished expression of COX-1 as well as COX-2 in the gallbladder mucosal epithelial cells. A sharp contrast was noted in our gallbladder mucocele and chronic cholecystitis reference groups, in which most of the cases showed positive immunoreaction to COX-1 and COX-2 in gallbladder mucosal epithelial cells. These results may suggest that canine gallbladder erosion/ulcer is related to a breakdown of local homeostasis as well as a decreased inflammatory reaction in the face of chronic inflammation. In other words, gallbladder erosion/ulcer can be a sign of a too-severe counter-reaction of the dog’s gallbladder to a long-standing irritable inflammation, so these topics should be the focus of future investigations.

Of importance, the paucity of control gallbladder specimens in the present study limits the interpretation of the results of histochemistry and IHC and their value. The authors possess many paraffin-embedded gallbladder specimens prepared from canine autopsy cases with no clinical history of gallbladder disease. These specimens, however, could not be used as clinical control specimens due to a significant loss of tinctorial features and cellular detail. The cause of the loss of these features is not precisely described in the literature but it is well known as “bile imbibition” and happens very commonly in the gallbladders in which formalin fixation was delayed. A well-planned tissue collection with a prompt formalin fixation should be exercised in future investigations. With this said, the results of various histochemistry and IHC of the clinical control dog in the current study should be reviewed with caution. For example, the negative results of the COX-1 IHC in the clinical control dog’s gallbladder was rather counter intuitive and might have reflected a mild background inflammation.

*Helicobacter pylori*, known as the bacterial cause of human gastric ulcers and malignancy, also increases the risk of chronic cholecystitis and cholelithiasis in humans [40]. Human patients with *H. pylori* in the gallbladder had a higher prevalence of acid regurgitation symptoms, histories of chronic gastritis, gastric ulcer, duodenal ulcer, and higher gastric *H. pylori* presence when compared to patients without intracholecystic *H. pylori* infection [41]. In addition to the WS stain, IHC is a common method to detect *H. pylori* in formalin-fixed paraffin-embedded (FFPE) specimens in humans [42,43]. Readily available antibodies for human *H. pylori* have been utilized to investigate intralesional non-*H. pylori* helicobacters in canine gastric FFPE specimens [44,45]. To the authors’ knowledge, the presence of spirochetes or helicobacters has not been reported in canine gallbladders. In the present investigation, we detected numerous long rods instead of spirochetes within an intra-cystically accumulated gel-like mucin in one case of gallbladder mucocele (GM4). These rods reacted to anti-human-*H. pylori* antibodies by IHC, but we could not perform an ultrastructural study to further characterize the pathogens. Further investigation on a larger cohort of canine gallbladder diseases through a variety of bacterial detection methodology is warranted to elucidate the importance of spirochetes and Helicobacters in canine gallbladder diseases.

In the present study, the HID-AB pH 2.5 method revealed intra- and inter-cohort differences between the GU, GM, and CC cohorts in staining patterns of canine gallbladder mucosal epithelial cells. The HID-AB pH 2.5 method is generally used to classify the acidic chemotype of mucins into sialomucins and sulfomucins based on the difference in terminal sialic acid or sulfate subgroups on the oligosaccharide chain [46]. The lack of a consistent trend in the chemical property of mucus produced by mucosal epithelial cells of the gallbladders of the GU, GM, and CC cohorts indicates the need for further investigation before the significance of the mucus chemical properties can be established, though a very detailed investigation on this matter has been conducted by Kesimer and colleagues [9].

Because of the retrospective nature of this study, there were some inevitable disadvantages. First, the small number of cases precluded the statistical analysis of the results, especially regarding the trend/influence of breed, sex, and neutering status. Secondly, ultrasonographic evaluations were not performed by board-certified radiologists. Thirdly, there was a partial lack of clinical and clinicopathological information for some of the submitted cases. A fixed format for gathering clinicopathological values should be used in future investigations. Fourthly, we were not able to speciate the bacteria. In the future, we or other investigators should design a study to avoid these deficiencies to give even more insights on canine gallbladder erosion/ulcer.

## 5. Conclusions

Canine gallbladder erosion/ulcer is an under-reported condition which requires vigorous clinical intervention. Based on the clinicopathological information reported in this study in addition to the COX-1 and COX-2 IHC results, we suggest that canine gallbladder erosion/ulcer may be related to decreased cytoprotection physiologically provided by arachidonic acid, but which is decreased or absent due to reduced COX expression because of yet undetermined etiologies. Further studies to elucidate the underlying molecular biological mechanism for a change in mucus production would be of value to provide better treatments for gallbladder erosion/ulcer.

## Figures and Tables

**Figure 1 animals-13-03335-f001:**
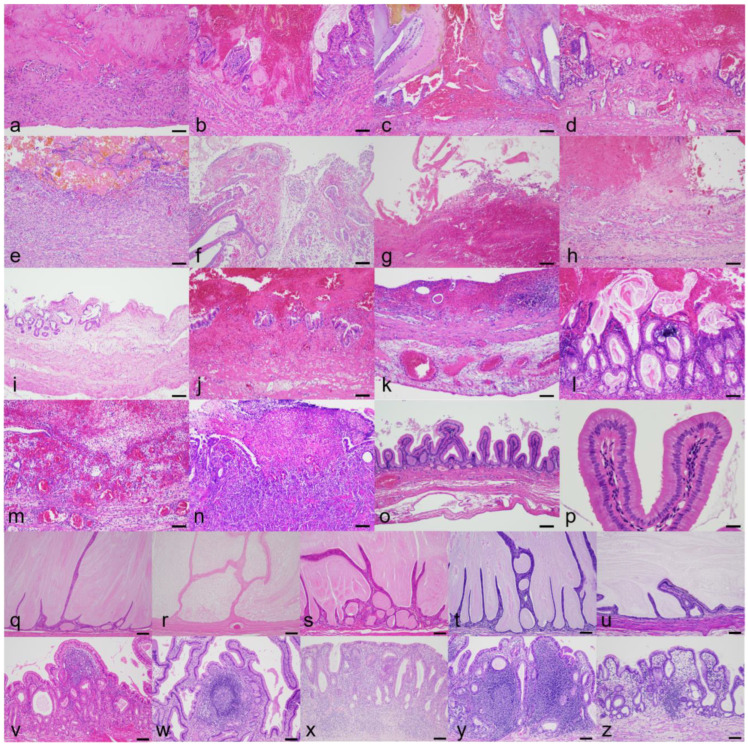
Overview of the representative histopathology of each disease cohort and the clinical control dog. Hematoxylin and eosin (H&E). (**a**–**n**) Gallbladder erosion/ulcer is accompanied by hemorrhage: (**a**) gallbladder erosion/ulcer case #1 (GU1); (**b**) GU2; (**c**) GU3; (**d**) GU4; (**e**) GU5; (**f**) GU6; (**g**) GU7; (**h**) GU8; (**i**) GU9; (**j**) GU10; (**k**) GU11; (**l**) GU12; (**m**) GU13; (**n**) GU14. (**o**) There are no significant lesions in the gallbladder of the clinical control dog. (**p**) Higher magnification of (**o**). (**q**–**u**) Gallbladder mucocele is characterized by abundant thick mucus accumulation in the lumen: (**q**) gallbladder mucocele case #1 (GM1); (**r**) GM2; (**s**) GM3; (**t**) GM4; (**u**) GM5. (**v**–**z**) Chronic cholecystitis manifests significant lymphoplasmacytic infiltrates in the lamina propria: (**v**) chronic cholecystitis case #1 (CC1); (**w**) CC2; (**x**) CC3; (**y**) CC4; (**z**) CC5. Bar = 100 µm except for (**p**), whose bar is 25 µm.

**Figure 2 animals-13-03335-f002:**
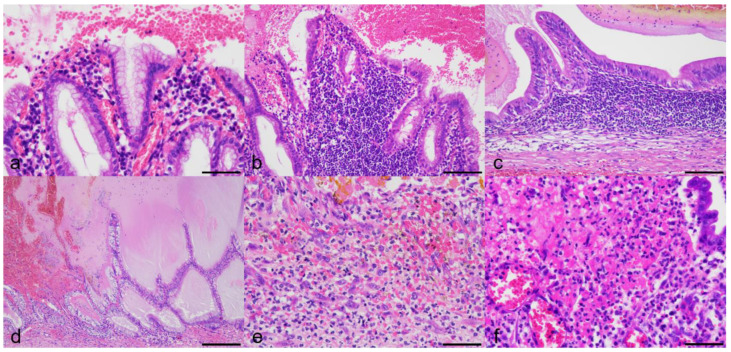
Representative histopathology of lesions of gallbladder erosion/ulcer. H&E. (**a**) Gallbladder mucosal epithelial cells at the rim of erosion/ulceration show no significant morphological change but rare pyknosis. GU4. Bar = 50 µm. (**b**) Marked lymphoplasmacytic infiltration in the lamina propria of the hyperplastic gallbladder mucosa. GU4. Bar = 100 µm. (**c**) Marked lymphoplasmacytic infiltration in the gallbladder mucosa. GU11. Bar = 100 µm. (**d**) Mucosal hyperplasia with accumulation of thick mucus in the gallbladder lumen (gallbladder mucocele). GU3. Bar = 200 µm. (**e**) Severe infiltration of neutrophils in the ulcerated mucosa. GU5. Bar = 50 µm. (**f**) Severe neutrophilic inflammation with hemorrhage and fibrin exudation in the ulcerated mucosa. GU14. Bar = 50 µm.

**Figure 3 animals-13-03335-f003:**
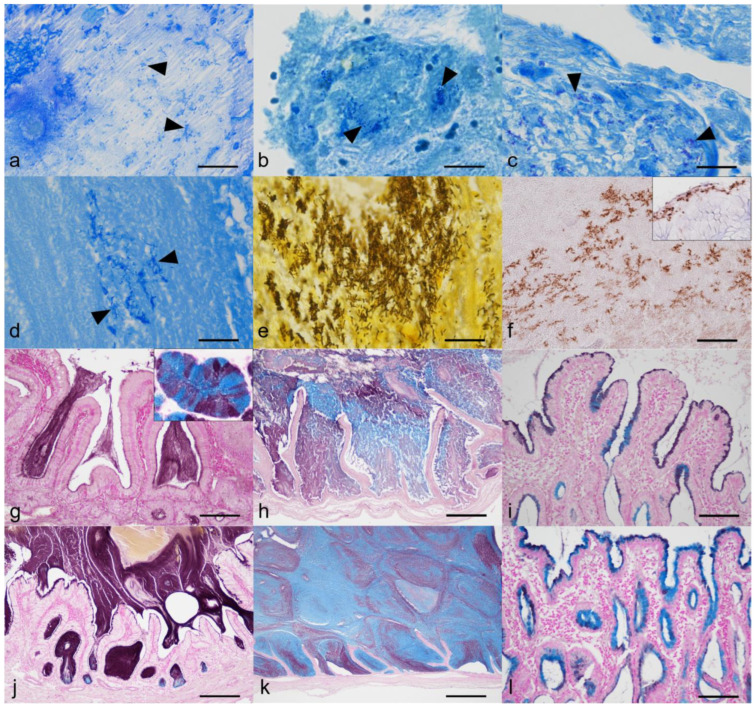
Results of bacterial detection and HID-AB pH2.5 staining pattern. (**a**) Cocci (arrowheads) in the gallbladder lumen. Giemsa. GU1. Bar = 20 µm. (**b**) Coccobacilli (arrowheads) in the gallbladder lumen. Giemsa. GU2. Bar = 20 µm. (**c**) Cocci (arrowheads) on the gallbladder mucosa. Giemsa. GU4. Bar = 20 µm. (**d**) Rods (arrowheads) in the gallbladder lumen. Giemsa. GM4. Bar = 20 µm. (**e**) Rods in the gallbladder lumen. WS. GM4. Bar = 20 µm. (**f**) Bacterial rods are immunopositive to anti-*Helicobacter* antibody. GM4. Bar = 50 µm. Inset: canine endoscopic gastric specimens with spirochetes for positive control. Immunohistochemistry for *Helicobacter*. (**g**) Mucus of the gallbladder of the clinical control dog shows sulfomucin-dominant pattern (black staining). HID-AB pH 2.5 method. Bar = 100 µm. Inset: esophageal gland of a dog as a positive control of HID-AB pH 2.5 method. (**h**) Gallbladder intraluminal mucus shows mixed staining pattern. HID-AB pH 2.5 method. GU1. Bar = 200 µm. (**i**) Gallbladder mucosal epithelial cells show two-tone staining pattern. HID-AB pH 2.5 method. GU4. Bar = 100 µm. (**j**) Gallbladder mucosal epithelial cells and mucus show sulfomucin-dominant pattern. HID-AB pH 2.5 method. GU2. Bar = 200 µm. (**k**) Gallbladder intraluminal mucus show sialomucin-dominant pattern. HID-AB pH 2.5 method. GM5. Bar = 500 µm. (**l**) Gallbladder mucosal epithelial cells show mixed staining pattern. HID-AB pH 2.5 method. CC5. Bar = 100 µm.

**Figure 4 animals-13-03335-f004:**
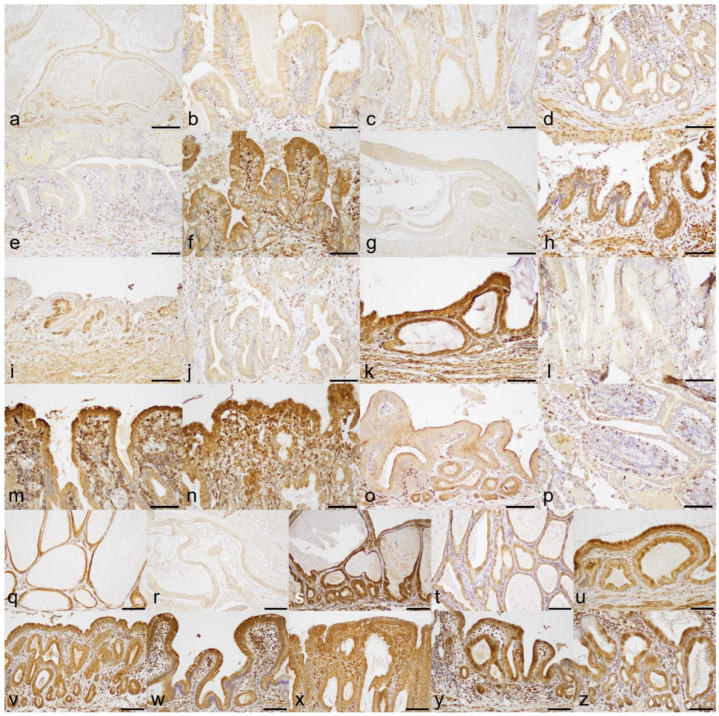
Results of anti-COX-1 immunohistochemistry: (**a**) gallbladder ulcer case #1 (GU1), negative; (**b**) GU2, negative; (**c**) GU3, negative; (**d**) GU4, negative; (**e**) GU5, negative; (**f**) GU6, positive; (**g**) GU7, negative; (**h**) GU8, positive; (**i**) GU9, negative; (**j**) GU10, negative; (**k**) GU11, positive; (**l**) GU12, negative; (**m**) GU13, positive; (**n**) GU14, positive; (**o**) gallbladder of the clinical control dog, negative; (**p**) tissue of canine small intestine as a positive control for COX-1 IHC, positive; (**q**) gallbladder mucocele case #1 (GM1), positive; (**r**) GM2, negative; (**s**) GM3, positive; (**t**) GM4, positive; (**u**) GM5, positive; (**v**) chronic cholecystitis case #1 (CC1), positive; (**w**) CC2, positive; (**x**) CC3, positive; (**y**) CC4, positive; (**z**) CC5, positive. Bar = 100 µm.

**Figure 5 animals-13-03335-f005:**
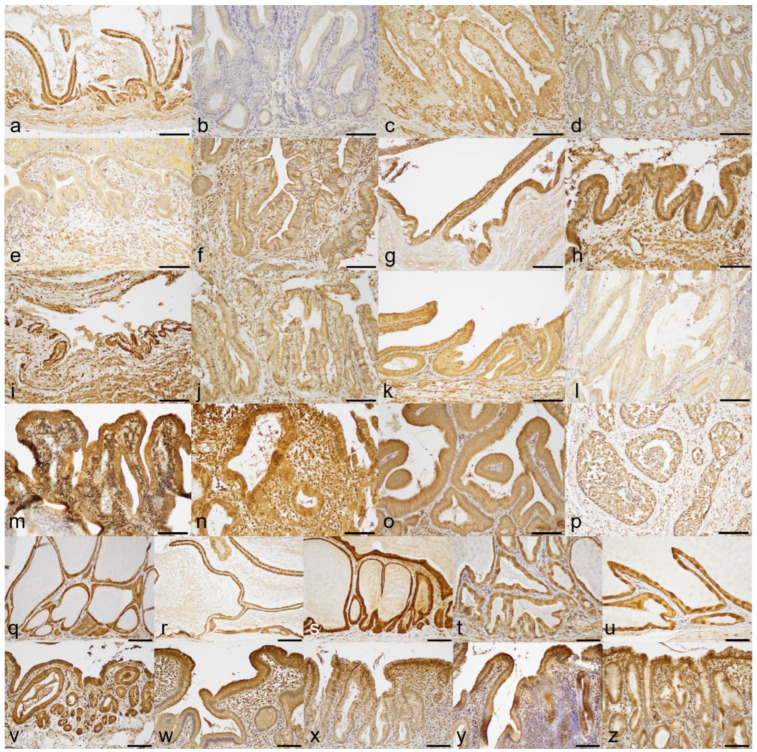
Results of anti-COX-2 immunohistochemistry: (**a**) gallbladder ulcer case #1 (GU1), positive; (**b**) GU2, negative; (**c**) GU3, negative; (**d**) GU4, negative; (**e**) GU5, negative; (**f**) GU6, positive; (**g**) GU7, positive; (**h**) GU8, negative; (**i**) GU9, negative; (**j**) GU10, negative; (**k**) GU11, negative; (**l**) GU12, negative; (**m**) GU13, positive; (**n**) GU14, positive; (**o**) gallbladder of the clinical control dog, negative; (**p**) tissue of canine mammary comedocarcinoma as positive control for COX-2 IHC, only tumor cells react to COX-2 antibody; (**q**) gallbladder mucocele case #1 (GM1), positive; (**r**) GM2, positive; (**s**) GM3, positive; (**t**) GM4, positive; (**u**) GM5, positive; (**v**) chronic cholecystitis case #1 (CC1), positive; (**w**) CC2, positive; (**x**) CC3, positive; (**y**) CC4, positive; (**z**) CC5, positive. Bar = 100 µm.

**Table 1 animals-13-03335-t001:** Primary antibodies for immunohistochemistry.

Antibody	Host	Type	Dilution	Source	Catalog Number
COX-1	Rabbit	Polyclonal	1:200	Proteintech Group, Inc., Rosemont, IL, USA	13393-1-AP
COX-2	Rabbit	Polyclonal	1:200	Proteintech Group, Inc., Rosemont, IL, USA	12375-1-AP
*Helicobacter*	Rabbit	Polyclonal	RTU ^a^	Nichirei Biosciences Inc., Tokyo, Japan	413151

^a^ Ready to use.

**Table 2 animals-13-03335-t002:** Results of histochemistry and immunohistochemistry.

Cases	WS ^d^	Giemsa	HID-AB ^h^ pH2.5	COX-1	COX-2	*Helicobacter*
**GU ^a^ 1**		C ^f^	Mixed		P	
**GU2**		C, CB ^g^	Su ^i^			
**GU3**			Mixed			
**GU4**		C, CB	Two-tone ^j^			
**GU5**		R, CB	Si ^k^			
**GU6**		C, CB	Su	P ^l^	P	
**GU7**			Su		P	
**GU8**			Su	P		
**GU9**			Unstained			
**GU10**			Su			
**GU11**			Mixed	P		
**GU12**		C	Mixed			
**GU13**		C	Mixed	P	P	
**GU14**		C	Two-tone	P	P	
**GM ^b^ 1**		CB	Mixed	P	P	
**GM2**		C, CB	Su		P	
**GM3**		C, CB	Mixed	P	P	
**GM4**	R ^e^	R	Su	P	P	P
**GM5**		C, CB	Si	P	P	
**CC ^c^ 1**			Su	P	P	
**CC2**			Su	P	P	
**CC3**			Mixed	P	P	
**CC4**			Mixed	P	P	
**CC5**			Mixed	P	P	
**Clinical control**			Su			

^a^ Gallbladder ulcer, ^b^ gallbladder mucocele, ^c^ chronic cholecystitis, ^d^ Warthin–Starry, ^e^ rods, ^f^ cocci, ^g^ coccobacilli, ^h^ high iron diamine-alcian blue, ^i^ sulfomucin-dominant, ^j^ sulfomucin by mucosal surface and sialomucin by crypts, ^k^ sialomucin-dominant, ^l^ immunopositive.

## Data Availability

The data presented in this study are available on request from the corresponding author. Caution should be exercised, however, that the whole-slide histology images (so-called virtual slides) are too heavy to transmit via email or other electronic transmission modalities. Requesters for these data should provide an HDD of at least 1TB.

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
