# Peer review of "Canine Gallbladder Erosion/Ulcer and Hemocholecyst: Clinicopathological Characteristics of 14 Cases"

_animals, 2023, doi:10.3390/ani13213335_

Round 1

Reviewer 1 Report

Comments and Suggestions for Authors

This paper, entitled

"Canine gallbladder erosion/ulcer and hemocholecyst: clinico-pathological characteristics of 14 cases with comparison to 3 gallbladder mucocele and chronic cholecystitis", is generally well laid out. Although it was carried out in a few cases, it addresses an important issue that is not widely explored in the veterinary scientific community. However, it leads to a meaningful discussion about the topics, approaches it comparatively with other diseases, and alerts the scientific community to an underdiagnosed disease.

One of the most significant limitations of the study is the absence of gallbladder controls without pathological alterations. I recognise that it is difficult to obtain them because often, when the order for necropsy is placed, there are already alterations in the mucosa that make it difficult to interpret the stains and immunohistochemistry. Despite this, this work should be valued, and in future work, when compared with the controls, perhaps new interpretations will emerge. So, I believe the work should be accepted for publication because the data may be essential for a starting point that may even contribute to new therapeutic approaches.

However, I have a few doubts:

- The title is too long, which makes it challenging to understand the article. Could it be shortened?

- The abstract has sentences similar to those in the text (e.g. Conclusions). It should be changed

2.2 - How was the macroscopic section cut? As measurements are made, this description is essential.

2.4 - Why do you say, "Slides were baked at 60°C for 30 160 minutes then deparaffinized"? What is the melting point of paraffin (not to be included in the text, but I would like clarification)? Why didn't you use a newborn dog kidney as a positive control for Cox-2? Not all breast tumours of the histological type described are Cox-2 positive.

Was Cox-2 positivity considered only in terms of extent, regardless of intensity? Why was this method chosen?

Are the material and method items and the results different, making it difficult to understand the article's logic? Can we please achieve more excellent uniformity? Perhaps include 3.1 and 3.2 in a more oversized item, for example, Clinical findings or description of clinical cases? And then subdivide 2m 3.1.1 and 3.1.2

Some histopathology photos (Figure 1) are blurry, and others have low contrast. Could these be improvements?

I recognise the technical difficulty of cutting the sample, but would it be possible to replace Figure 3 K with another with fewer striations?

 The immunocytochemistry figures seem to have little contrast as blue. Can this be improved?

Reviewer 2 Report

Comments and Suggestions for Authors

1. The proportion is relatively small, the whether sterilization has an effect on the disease is not significant.

2. There is no bar word describsion of proportion of magnification in all histopathological images.

Reviewer 3 Report

Comments and Suggestions for Authors The authors explore retrospective cases of canine gallbladder erosion/ulcer to further characterize this pathology.   1. The number of cases is quite limitative and this should be addressed, especially when breed, sex and castrated/spayed considerations are stated on the manuscript. The lack of healthy controls is also concerning. 2. The authors also described animals were submitted to CBC and blood chemistry but it is not possible to know which panel was used for those screenings. A detailed description of what was considered should be at the manuscript and absolute values for each animal could be added as supplementary material. 3. Inflammatory infiltrates are not described in an accurate and quantitative way. This should be improved as it could lead to further IHC characterization. 4. Was an Helicobacter PCR on fixed tissue attempted? Comments on the Quality of English Language

An English revision is required throughout the text.
